# Optimizing ECG to detect echocardiographic left ventricular hypertrophy with computer-based ECG data and machine learning

**Fernando De la Garza Salazar**[1,2], **Maria Elena Romero Ibarguengoitia**[1,3], **José Ramón Azpiri López**[4], **Arnulfo González Cantú**[1,3]*

**1** School of Medicine, Medical Specialties, University of Monterrey, Monterrey, Nuevo León, Mexico,
**2** Department of Internal Medicine, *Hospital Christus Muguerza Alta Especialidad*, Monterrey, Nuevo León, Mexico, **3** Department of Medical Education and Research in Health, *Christus Muguerza* Health Systems, Monterrey, Nuevo León, Mexico, **4** Department of Cardiology, *Hospital Christus Muguerza*, *Alta Especialidad*, Monterrey, Nuevo León, Mexico

* drgzzcantu@gmail.com

## Abstract

### Background

Left ventricular hypertrophy detected by echocardiography (Echo-LVH) is an independent predictor of mortality. Integration of the Philips DXL-16 algorithm into the electrocardiogram (ECG) extensively analyses the electricity of the heart. Machine learning techniques such as the C5.0 could lead to a new decision tree criterion to detect Echo-LVH.

### Objectives

To search for a new combination of ECG parameters predictive of Echo-LVH. The final model is called the Cardiac Hypertrophy Computer-based model (CHCM).

### Methods

We extracted the 458 ECG parameters provided by the Philips DXL-16 algorithm in patients with Echo-LVH and controls. We used the C5.0 ML algorithm to train, test, and validate the CHCM. We compared its diagnostic performance to validate state-of-the-art criteria in our patient cohort.

### Results

We included 439 patients and considered an alpha value of 0.05 and a power of 99%. The CHCM includes T voltage in I ($\leq$0.055 mV), peak-to-peak QRS distance in aVL (>1.235 mV), and peak-to-peak QRS distance in aVF (>0.178 mV). The CHCM had an accuracy of 70.5% (CI95%, 65.2–75.5), a sensitivity of 74.3%, and a specificity of 68.7%. In the external validation cohort (n = 156), the CHCM had an accuracy of 63.5% (CI95%, 55.4–71), a sensitivity of 42%, and a specificity of 82.9%. The accuracies of the most relevant state-of-the-art criteria were: Romhilt-Estes (57.4%, CI95% 49–65.5), VDP Cornell (55.7%, CI95%47.6–63.7),

**Data Availability Statement:** All relevant data are within the manuscript and its Supporting information files.

**Funding:** The author(s) received no specific funding for this work.

**Competing interests:** The authors have declared that no competing interests exist.

Cornell (59%, CI95%50.8–66.8), Dalfó (62.9%, CI95%54.7–70.6), Sokolow Lyon (53.9%, CI95%45.7–61.9), and Philips DXL-16 algorithm (54.5%, CI95%46.3–62.5).

## Conclusion

ECG computer-based data and the C5.0 determined a new set of ECG parameters to predict Echo-LVH. The CHCM classifies patients as Echo-LVH with repolarization abnormalities or LVH with increased voltage. The CHCM has a similar accuracy, and is slightly more sensitive than the state-of-the-art criteria.

## Introduction

Left ventricular hypertrophy (LVH) is a strong predictor of cardiovascular morbidity and mortality [1–8]. Echocardiography is an excellent tool to detect LVH (Echo-LVH), but it is not widely available. The electrocardiogram (ECG) is a low-cost tool, and is frequently used to predict Echo-LVH; however, low accuracy and sensitivity have historically limited its true potential [1, 9–24]. Current guidelines recommend dozens of ECG criteria to detect LVH with no preference of one over another, and include: limb-lead voltage criteria (i.e., R aVL >11mm Sokolow), precordial lead-voltage criteria (i.e., S V1 >23mm Wilson), a combination of limb and precordial voltage criteria (i.e., Total 12-lead voltage >175mm Siegel), and a combination of voltage and non-voltage criteria (i.e., Romhilt-Estes) [25]. We previously calculated 22 ECG criteria and demonstrated that the Dalfó criteria (SV3 + RaVL) had the overall best accuracy in the detection of Echo-LVH in our population and in specific populations (i.e., hypertensives), with an accuracy of 64.1%, a sensitivity of 56%, and a specificity of 71.3% [24]. Therefore, we have begun to use other techniques to increase the diagnostic performance of the ECG in the detection of Echo-LVH.

Advanced statistical techniques such as Bayesian regression trees, neural networks, and the C5.0 machine learning algorithm (C5.0) have previously optimized the ECG to predict Echo-LVH [22, 23, 26]. Bayesian regression trees and neural networks have reported accuracy, sensitivities, and specificities in the order of 80%, 40%, and 90%, respectively [22, 23]. However, these models are black-box models, and the clinician is unaware of which ECG abnormality identified the patient as normal or as Echo-LVH. The black-box nature of Bayesian regression trees and neural networks limits its clinical applicability. In contrast, the C5.0 ML algorithm is a non-black-box model that creates a decision tree structure that is easy to understand and implement. The C5.0 automatically searches for the best ECG parameters in the tree branches, creating "paths" or ECG phenotypes [26]. We previously developed the decision tree "Cardiac hypertrophy model" that detects three distinct Echo-LVH phenotypes: ST ECG-LVH, voltage-left atrial enlargement LVH, and voltage-duration LVH, predicting Echo-LVH with an accuracy of 73.3%, a sensitivity of 81.6%, and a specificity of 69.3% [26]. However, we used a small number of ECG parameters obtained by manual readings [26]. Manual readings are associated with human cognitive bias and high inter- and intra-rate variability. Also, humans neglect minor electrical variations that could potentially predict Echo-LVH [26].

On the other hand, more advanced computer-based ECG software (i.e., Philips DXL-16 algorithm) allow an extensive analysis of the heart's electricity. The Philips DXL-16 algorithm refers to software integrated into the digital ECG. It offers 458 ECG parameters per ECG recording, including standard (i.e., S and R wave amplitude and duration), and other

non-standard ECG parameters (i.e., ST area, T wave area). No studies have explored the predictive capacity of many of these ECG parameters.

We hypothesized that the Philips DXL-16 and C5.0 could find a new combination of ECG parameters to predict Echo-LVH. This paper refers to the training, testing, and validation of the cardiac hypertrophy computer-based model (CHCM). Also, we analyzed and interpreted the final model based on its phenotype physiopathology. Finally, we compared the diagnostic performance of the CHCM with the most relevant state-of-the-art criteria in our population.

## Patients and methods

### Study designs and settings

The local ethics committee approved this study before data collection (CMHAE-001-19). This study also complies with the latest revision of the Declaration of Helsinki. We protected the confidentiality and privacy of the research subjects. The local ethics committee waived the requirement of an informed consent form due to the study´s retrospective nature.

This was an observational, retrospective, case-control study. Training/testing and validation sets were obtained retrospectively in consecutive adults who underwent an echocardiogram (Echo) and an ECG (with less than seven days difference between studies), between January 2016 and May 2019 in the Cardiology Department of the *Hospital Christus Muguerza Alta Especialidad* in Monterrey, Mexico. We followed the STARD methodology to report diagnostic accuracy [27], and international guidelines for the development of Machine Learning (ML) models [28].

### Sample of patients

A total of 5650 consecutive patients underwent an Echo at the time of the study. Initially, we excluded patients by age (<35 years, n = 2150), and those lacking Echo LV measurements or an ECG (n = 2,185). We subsequently applied the other exclusion criteria in 270 patients: fusion or pacemaker rhythms, preexcitation syndromes, and intraventricular conduction delays (bundle branch block and hemiblock), hypertrophic cardiomyopathy, dilated cardiomyopathy, interventricular septal defects, acute myocardial ischemia, elevated cardiac enzymes, tachycardia (>110 bpm), ICU patients, prior cardiotomy (<3 months). Patients with IHD were accepted since this disease is very common in patients with Echo-LVH. Echo and ECG findings compatible with ischemia included segmental akinesia or hypokinesia of the cardiac vascularized territory irrespective of the presence of pathological Q waves in two or more contiguous leads.

The initial sample of 439 patients was split into two groups: training (70%, n = 307) and testing set (30%, n = 132). The external validation sample included 156 patients.

### Data sources

**Participant data.** Demographic data, such as gender, age, height (cm), weight (kg), and medical comorbidities (i.e., hypertension, type 2 diabetes mellitus), were collected from medical records. We reported the body mass index in kg/ $m^2$, and calculated the body surface area with Mosteller's formula as follows: BSA = $\sqrt{}$ (weight * height) / 3600.

**Echocardiographic data.** All transthoracic Echo were obtained following a two-dimensional ECG guided M mode approach, and according to the European Association of Cardiovascular Imaging and the American Society of Echocardiography guidelines and recommendations. Three licensed cardiologists performed a transthoracic Echo using the

"EPIQ7" and "IE33" Phillips (Best, Netherlands) equipment (agreement kappa = 0.91) (29). We obtained the following ventricular measurements in diastole:

- Left ventricular internal diameter (LVID)

- Interventricular septum thickness (IVST)

- Left ventricular posterior wall thickness (LVPWT)

- Left ventricular mass (LVM, gr)

LVM was calculated with the following formula [29]:

$$LVM = 0.8 \times \left\{ 1.04 \left[ (LVID + LVPWT + IVST)^3 - (LVID)^3 \right] \right\} + 0.6g$$

The "American Society of Echocardiography and the European Association of Cardiovascular Imaging" recommend indexation of the LVM with BSA [29]. We calculated the LMVI (gr/ $m^2$) as follows:

$$LVMI = LVM \div BSA$$

We defined LVH as: male patients with a LVMI above 115 gr/$m^2$, and above 95 gr/$m^2$ in female patients [29]. Severity was defined as mild (male: 116–131 gr/$m^2$, female: 96–108 gr/$m^2$), moderate (male: 132–148 gr/$m^2$, female: 109–121 gr/$m^2$), and severe (male: >148 gr/$m^2$, female: >121 gr/$m^2$) [29].

Different left ventricular geometries were defined according to the relative wall thickness (RWT) ($RWT = (2 \times LVPWT) \div LVID$)), as cardiac remodeling (normal LVMI and RWT >0.42), concentric hypertrophy (elevated LVMI and RWT >0.42), and eccentric hypertrophy (elevated LVMI and RWT ≤0.42) [29].

**ECG data—Philips DXL-16 algorithm.** All 12-lead ECGs were performed using a 25mm/sec velocity and 10 mm/mV sensitivity. We used the Philips "Pagewriter TC50" equipment (Best, The Netherlands), with the incorporated Philips DXL-16 algorithm. The Philips DXL-16 algorithm measures 458 ECG parameters in each ECG recording. These ECG parameters were extracted using the program PDF element 6 Pro 14.40.26. All values were corroborated directly. These ECG measurements include standard (i.e., R voltage, R duration), and non-standard parameters (i.e., QRS PPK-QRS, ST area). The S1 Table shows all the ECG parameters obtained with the Philips DXL-16 algorithm. For more information on the DXL-16 software, please consult the owners' guide PDF.

## ECG criteria comparisons

We previously calculated 22 ECG criteria for Echo-LVH in the sample of patients included in this study; the best criteria were recalculated and compared to the CHCM. The recalculated criteria were selected on the basis of accuracy (i.e., Dalfó), common use (i.e., Cornell, Sokolow), and complexity (i.e., Voltage Duration Product [VDP] Cornell, Philips DXL-16 algorithm, and Romhilt-Estes) [24]. The selected criteria were calculated as follows [16, 25, 30, 31]: 1) Sokolow Lyon: R aVL >1.1 mV; 2) Cornell: S V3 + R aVL male: > 2.8mV or female: > 2.0mV; 3) Dalfó S V3 + R aVL male: > 1.6mV or female: > 1.4mV; 4) Romhilt-Estes: negative P terminal force ≥1 Ashman unit (3 points); R or S wave in any limb lead ≥2 mV, or R wave in V5 or V6 ≥3 mV (3 points), or S wave in V1 or V2 ≥3 mV; ST "strain" pattern = downward ST depression >1 mm at 40ms from the J point with a downward slope and with asymmetric T wave inversion, without digitalis (3 points); QRS duration ≥ 0.09 msec [1 point]; left axis deviation defined as QRS axis ≤ −30 degrees [2 points]; Intrinsicoid deflection in V5 or

V6 ≥ 0.05 msec [1 point], and scored LVH ≥ 4 points; 5) VDP Cornell: (RaVL + SV3 + 0.6mV) x QRS duration (female) or VDP (RaVL + SV3) x QRS duration (male) > 244 mV*msec. We ignored less accurate ECG criteria (i.e., Sokolow-Lyon index, S in V1 + R in V5 or V6 (whichever is greater) {greater than or equal to} 35 mm ({greater than or equal to} 7 large squares).

The Philips DXL-16 algorithm also includes a model that detects Echo-LVH based on a point score derived from several findings: high voltage in QRS components, left axis deviation in the frontal plane, left atrial enlargement, ST-T changes characteristic of LVH, and a prolonged QRS duration of the ventricular activation time. Higher scores result in a greater likelihood of LVH.

We also calculated the *Cardiac hypertrophy model* -our previous model- created using ECG manual readings and the C5.0 ML algorithm [26].

## C5.0 machine learning algorithm and creation of the CHCM

All the Philips DXL-16 ECG parameters were fed into the C5.0 ML algorithm to create the CHCM. The C5.0 ML algorithm is a ML method that defines a decision tree structure model (or criteria). The decision trees are easy to understand and implement. The tree model comprises a series of logical decisions, similar to a flow chart, with decisions split into branches that indicate the decision choice. This decision-making process ultimately leads to a leaf node (or terminal node) that denotes the combined decisions. The terminal node reflects the probability of each individual to be classified in a predicted class (i.e., positive or negative for Echo-LVH). If the probability of being classified as positive for Echo-LVH is superior to 50%, the criteria will be positive and vice-versa. [32].

The C5.0 model automatically selects the best set of ECG parameters to detect Echo-LVH based on *information gain*. The C5.0 algorithm uses entropy to calculate the change in homogeneity at each split using different predictors (i.e., all 458 ECG standard and non-standard parameters per patient). This is how we finally select the final set of ECG parameters with their respective branching cut-off values [32].

To prioritize sensitivity rather than specificity, we created a matrix cost that penalizes false negatives more than false positives. Also, decision trees can continue to grow indefinitely -this is called overfitting- and curbs generalization of new information.

The C5.0 algorithm created a global pruning stage to avoid overfitting; the C5.0 takes the tree as a whole and collapses weak subtrees.

## Statistical analysis

Normality was established by applying the Shapiro-Wilk test, and log10 transformations were conducted when appropriate. We described continuous variables as mean and standard deviation or confidence intervals, and categorical variables as frequencies and percentages. We compared groups using a two-sample t-test and Fisher's exact test.

We calculated and compared the diagnostic utility/performance (accuracy CI95%, sensitivity, specificity, positive and negative predictive value) of all the included ECG criteria (the CHCM, our previous "Cardiac hypertrophy model," the Philips DXL-16 algorithm, Romhilt-Estes, VDP Cornell, Cornell, Dalfó, and Sokolow-Lyon).

## Sample size calculation and missing values

The sample size was calculated based on a sensitivity of 40% of the reported conventional electrocardiographic criteria, using a delta sensitivity of 0.1 (inferiority sensitivity limit = 30%) [25]. We required at least 155 patients per group to reach a power of 80% and an alpha error

<0.05. No missing values in demographic, Echo and ECG parameters were accepted, although complete case analyses that included comorbidities were acceptable. The models were two-sided, and the significant p-value was <0.05. We used the statistics program SPSS vs. 24 and R studio vs. 3.4.0.

## Results

### Description of patient demographics, anthropometric values, Echo measurements, and comorbidities

With the total sample, the study reached a power of 99%. The first sample was split into two groups: 70% (n = 307) for training, and another 30% for testing (n = 132) of the CHCM. A second sample was included for external validation [156, 35.5%]).

The training/testing set included a total of 439 patients. Two hundred three (46.2%) patients had Echo LVH, and 236 were controls. Of these 439 patients, 104 had Echo positive ischemic findings (48 Echo LVH). Patients in the training/testing set that were positive for Echo-LVH tended to be older and male (Table 1). This difference was only true in the sub-group of patients with IHD (p = 0.001). Demographic, anthropometric, Echo measurements, and comorbidity comparisons between patients with Echo-LVH and controls in the training/testing set are shown in Tables 1 and 2. The prevalence of atrial fibrillation, chronic heart failure, aortic stenosis, hypertension, and hypothyroidism was different between cases and controls (p<0.05). All Echo measurements were statistically different between patients with Echo-LVH and controls (p<0.01) (Table 1). The distribution of left ventricular morphology patterns was: normal morphology (n = 100, 22.8%), cardiac remodeling (n = 136, 31%), concentric LVH (n = 166, 37.8%), and eccentric LVH (n = 37, 8.4%). LVH severity stage was categorized as: mild (n = 76, 37.4%), moderate (n = 50, 24.6%), and severe (n = 77, 37.9%). The severity stages of LVH between males and females were similar (p = 0.420).

**Table 1. Demographic and echocardiographic measurements of the population.**

| Mean (SD) | Total sample (n = 439, 100%) | Echo | | p-value |
|---|---|---|---|---|
| | | Negative LVH (n = 236, 53.8%) | Positive LVH (n = 203, 46.2%) | |
| Demographic and Anthropometric characteristics | | | | |
| Age (years) | 67.2 (13.5) | 65.8 (14.5) | 68.8 (12.1) | 0.02 |
| Weight (kg) | 78 (16.8) | 77.8 (16.1) | 78.2 (17.6) | 0.77 |
| Height (cm) | 167.2 (9.7) | 168.1 (9.6) | 166.3 (9.7) | 0.07 |
| BMI (kg/m$^2$) | 27.8 (4.9) | 27.3 (4.4) | 28.2 (5.4) | 0.12 |
| BSA (m$^2$) | 1.9 (0.24) | 1.9 (0.23) | 1.9 (0.25) | 0.84 |
| Echo parameters | | | | |
| IVST | 1.18 (0.27) | 1.04 (0.19) | 1.34 (0.27) | 0.001 |
| LVID | 4.56 (0.75) | 4.38 (0.63) | 4.77 (0.8) | 0.001 |
| LVPWT | 1.18 (0.26) | 1.04 (0.19) | 1.33 (0.24) | 0.001 |
| RWT | 0.54 (0.2) | 0.49 (0.16) | 0.58 (0.2) | 0.001 |
| LVM (gr) | 201.6 (73.8) | 155.6 (38) | 255 (69.3) | 0.001 |
| LVMI (gr/m$^2$) | 105.9 (34.9) | 81.6 (15.5) | 134.2 (29.4) | 0.001 |

Demographics of patients with Echo LVH vs. controls. Both groups were overweight and had similar body surface areas. Age was similar in all patients except in those with Echo-IHD.

Abbreviations: BMI: body mass index; BSA: body surface area; Echo: echocardiogram, IVST: interventricular septum thickness diastole; LVH: left ventricular hypertrophy; LVID: left ventricular internal diameter diastole, LVPWT: left ventricular posterior wall thickness diastole, LVM: left ventricular mass, LVMI: left ventricular mass index, RWT: relative wall thickness.

**Table 2. Population comorbidities.**

| | Total Samples (n = 321, 100%) | Negative ECHO-LVH (n = 145, 45.1%) | Positive ECHO-LVH (n = 176, 54.8%) | p-value |
|---|---|---|---|---|
| **AF** | 63 (19.6) | 19 (13.1) | 44 (25) | 0.011 |
| **Aortic stenosis** | 17 (5.3) | 1 (0.7) | 16 (9.1) | 0.001 |
| **IHD** | 171 (52.3) | 84 (53.2) | 87 (51.5) | 0.825 |
| **CHF** | 44 (13.7) | 13 (9) | 31 (17.6) | 0.03 |
| **CKD** | 32 (10) | 9 (6.2) | 23 (13.1) | 0.06 |
| **COPD** | 9 (2.8) | 1 (0.7) | 8 (4.5) | 0.044 |
| **Dyslipidemia** | 60 (18.7) | 23 (15.9) | 37 (21) | 0.253 |
| **DM2** | 114 (35.5) | 43 (29.7) | 71 (40.3) | 0.061 |
| **Hypertension** | 207 (64.4) | 81 (25.2) | 126 (39.2) | 0.001 |
| **Hypothyroidism** | 33 (10.3) | 9 (6.2) | 24 (13.6) | 0.041 |
| **OSA** | 3 (0.9) | 1 (0.7) | 2 (1.1) | 1.0 |
| **PAD** | 15 (4.7) | 8 (5.5) | 7 (4) | 0.599 |
| **PH** | 5 (1.6) | 2 (1.4) | 3 (1.7) | 1.0 |
| **PE** | 7 (2.2) | 4 (2.8) | 3 (1.7) | 0.705 |
| **SSS** | 3 (0.9) | 1 (0.7) | 2 (1.1) | 1.0 |
| **Stroke** | 34 (10.6) | 13 (9) | 21 (11.9) | 0.467 |
| **SVT** | 9 (2.8) | 5 (3.4) | 4 (2.3) | 0.736 |

Completely missing random comorbidity values were 26.9% of the total sample. We performed complete case analyses.

Abbreviations: AF: atrial fibrillation, CHF: congestive heart failure, CKD: chronic kidney disease, COPD: chronic obstructive pulmonary disease, DM2: type 2 diabetes mellitus, IHD: ischemic heart disease, OSA: obstructive sleep apnea, PAD: peripheral artery disease, PE: pulmonary embolism, PH: pulmonary hypertension, SSS: sick sinus syndrome, SVT: supraventricular tachycardia.

The external validation set included 156 patients, 35.5% of the initial sample. In the external validation set, the prevalence of LVH was 47.4%. In this set, males were more prevalent (83, 53.2%) than females, the mean age was 64.3 years (13.9), and the overall BMI was 28.4 kg/m$^2$ (5.2). Gender and age were similar between patients with Echo-LVH and controls (p>0.05).

## CHCM diagnostic performance and external validation

In the testing set, the CHCM reached a diagnostic accuracy of 70.5% (CI95%, 65.2–75.5), a sensitivity of 74.3%, specificity 68.7%, PPV 53.8%, and NPV 84.5%. The *CHCM* includes three ECG parameters (three nodes) and has four levels. No isolated ECG parameter can classify patients as having Echo-LVH. The *CHCM* detects two electrical LVH phenotypes (Fig 1). In the external validation set, the CHCM reached a lower accuracy 63.5% (CI95%, 55.4–71), sensitivity (42%), PPV (68.9%), NPV (61.3%), but higher specificity (82.9%).

## CHCM vs. the most useful state-of-the-art-criteria in our population

Most state-of-the-art criteria have greater specificity than the CHCM. The CHCM showed higher accuracy and a similar NPV than the most accurate criterion in our population (Dalfó criteria) (Table 3). However, the CHCM included both depolarization and repolarization ECG abnormalities, and the Dalfó criteria are based solely on voltage, therefore excluding relevant information. The *CHCM* has more accuracy and NPV than the Cornell, VDP Cornell, and Sokolow-Lyon (Table 3). It is also valid for more than a dozen common criteria previously validated in our population [24]. Other more complex criteria (i.e., Romhilt-Estes, Philips DXL-16 algorithm) are also less accurate (Table 3). The accuracy and sensitivity of the CHCM

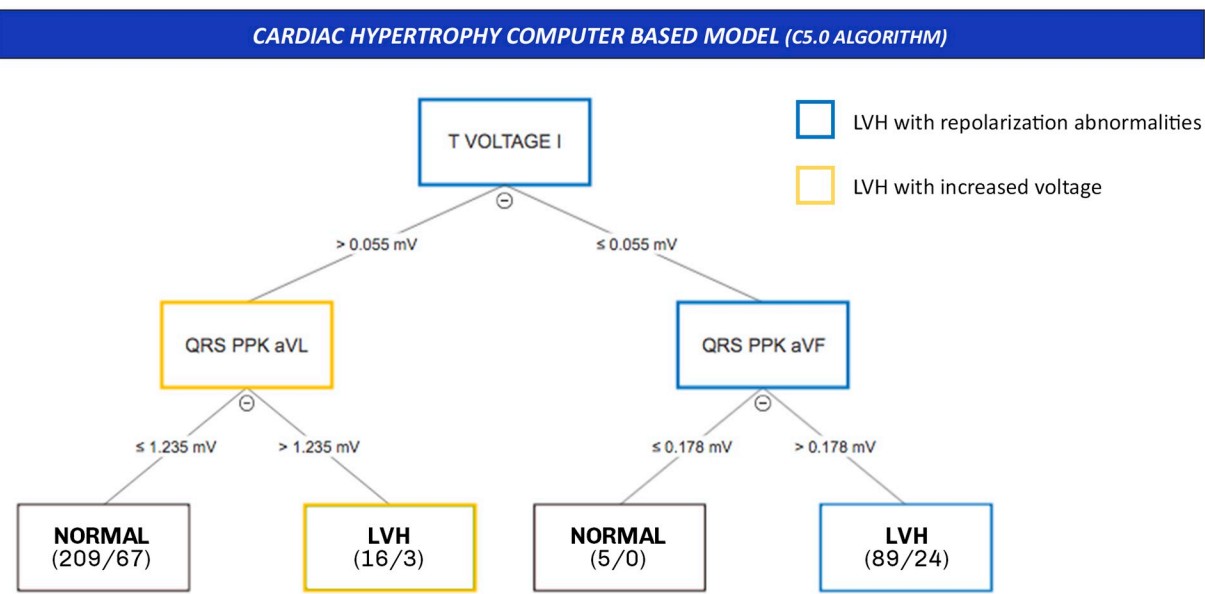

**Fig 1. Cardiac Hypertrophy Computer-based model (CHCM) and the electrical LVH phenotypes.** The CHCM included T wave voltage in lead I, and the peak-to-peak QRS distance (QRS PPK) in aVF and aVL. Note that no isolated electrical change can classify LVH. The CHCM describes two electrical LVH phenotypes: LVH with repolarization abnormalities, and LVH with increased voltage. Abbreviations: T voltage I: amplitude in mV of T wave amplitude in DI, QRS PPK, aVL: Peak-to-peak QRS complex amplitude in aVL; QRS PPK, aVF: Peak-to-peak QRS complex amplitude in aVF, LVH: left ventricular hypertrophy.

dropped in the external validation set from 70.5% to 63.5% and 74.3% to 42%, respectively (Table 3). The external validation set sensitivity is inferior to that in our previously published model (Cardiac hypertrophy model) [26]. However, the CHCM is simpler, more specific (82.9% vs. 69.3%), and has a greater PPV (68.9% vs. 56.3%) (Table 3).

**Table 3. Comparison of the diagnostic utility of standard ECG criteria and the CHCM.**

| Criteria | # of electrical variables | Delta ac (%) | Ac (%, CI95%) | Se (%) | Sp (%) | PPV (%) | NPV (%) |
|---|---|---|---|---|---|---|---|
| Cardiac hypertrophy computer-based model (CHCM) Testing set | 3 | reference | 70.5 (65.2–75.5) | 74.3 | 68.7 | 53.8 | 84.5 |
| Cardiac hypertrophy computer-based model (CHCM) External validation set | 3 | -7 | 63.5 (55.4–71) | 42 | 82.9 | 68.9 | 61.3 |
| Cardiac hypertrophy model | 6 | +2.8 | 73.3 (65.5–80.2) | 81.6 | 69.3 | 56.3 | 88.6 |
| Philips DXL-16 algorithm | >10 | -16 | 54.5 (46.3–62.5) | 6.8 | 97.6 | 71.4 | 53.7 |
| Romhilt Estes | 7 | -13.1 | 57.4 (49–65.5) | 11.6 | 97.4 | 80 | 55.8 |
| VDP Cornell | 3 | -14.8 | 55.7 (47.6–63.7) | 8.1 | 98.8 | 85.7 | 54.4 |
| Cornell | 2 | -11.5 | 59 (50.8–66.8) | 17.6 | 96.3 | 81.3 | 56.4 |
| Dalfó | 2 | -7.6 | 62.9 (54.7–70.6) | 61.1 | 64.6 | 61.1 | 64.6 |
| Sokolow Lyon | 1 | -16.6 | 53.9 (45.7–61.9) | 14.9 | 89 | 55 | 53.7 |

Table 3 shows the diagnostic utility comparisons of all the calculated criteria and the *CHCM*.

Abbreviations: Accuracy (Ac), negative predictive value (NPV), positive predictive value (PPV), sensitivity (Se), specificity (Sp), voltage duration product (VDP).

## Discussion

Computer-based ECG parameters (Philips DXL-16 algorithm) and C5.0 established a new combination of ECG parameters to detect Echo-LVH with a comparable diagnostic performance to the most useful state-of-the-art criteria in our population. This new decision tree criterion is called CHCM (Fig 1). The CHCM is easy to understand and includes ECG parameters not previously reported in any LVH criteria: T wave voltage in the lead I, peak-to-peak QRS distance (QRS PPK) in aVF, and peak-to-peak QRS distance in aVL [25]. The *CHCM* model differentiates two electrical LVH phenotypes: 1) LVH with repolarization abnormalities and 2) LVH with high voltage [7, 25] (Fig 1). This study explored the predictive capacity of 458 computer-based ECG parameters, whereby the C5.0 algorithm selected the best ECG parameters, their relative position in the tree, and their precise cut-off value.

We directly compared the CHCM accuracy, sensitivity, specificity, PPV, and NPV to other criteria: classical 22 state-of-the-art criteria, the Philips DXL-16 algorithm [point score system], and our previously published *Cardiac hypertrophy model* (decision tree criteria based on ECG manual readings) (Table 3). The CHCM has greater accuracy (70.5%), sensitivity (74.3%), and NPV (84.5%) than 22 of the state-of-the-art criteria in our population [24]. In our cohort of Mexican subjects, the sensitivity of most of the classic criteria is below 20% [24, 33]. The Dalfó voltage criteria [SV3 + RaVL >1.6mV male 1.4mV women]—a variation of the Cornell voltage criteria—is the most accurate classic ECG criterion in Mexicans, with an accuracy of 62.9% and well-balanced sensitivity (61.1%) and specificity (64.6%). The CHCM has greater accuracy (70.5% vs 62.9%), sensitivity (74.3% 61.1%), and slightly better specificity than the Dalfó criteria (68.7% vs 64.6%). The CHCM's multi-leveled nature could explain this difference.

Multilevel score systems such as the CHCM, Romhilt-Estes, Philips DXL-16, and the Cardiac hypertrophy model have advantages over criteria based solely on voltage. For example, using multiple ECG parameters (i.e., ST abnormalities, voltage, or duration changes) to predict Echo-LVH provides a more realistic model of the disease. However, most of the multi-level criteria currently used are inaccurate, mainly the result of low sensitivity. Romhilt-Estes had an accuracy of 57.4% and a sensitivity of 11.6%; the Philips DXL-16 algorithm had an accuracy of 54.5% and a sensitivity of 6.8% in this study (Table 3). Moreover, a recent retrospective study that included a large, diverse population of 13,960 patients with Echo-LVH and controls, found that the ECG software MUSE® 12-SL (GE Healthcare, Chicago, IL)– similar to the Philips DXL-16 algorithm- reported a sensitivity in LVH detection of 30.7% and a specificity of 84.4% [34]. The CHCM is more accurate and uses fewer ECG parameters/predictors [22–24, 26, 35]. It seems that some CHCM characteristics, such as taking into account minimal changes in T wave amplitude (i.e., T wave voltage in lead I <0.122 mV), and the absolute QRS amplitude (peak-to-peak distance of the QRS), are good ECG predictors of Echo-LVH [20, 36]. Unlike Romhilt-Estes and the Cardiac hypertrophy model, the CHCM does not use atrial information, but rather more subtle variations in repolarization than ST-strain. The ST-strain pattern is uncommon after the antihypertensive era, and previous evidence supports that sub-millimetric changes in the ST segment can predict Echo-LVH [5, 20]. The *CHCM* uses the T wave voltage in lead I as the best classifier of Echo-LVH, reflecting that abnormal repolarization is a major finding in Echo-LVH. Interestingly, none of the criteria recommended by international guidelines includes T wave abnormalities in any lead [25].

The CHCM has some advantages over classic ECG criteria (i.e., Cornell, Sokolov). For example, the CHCM uses depolarization and repolarization abnormalities. The

Atherosclerosis Risk in Communities (ARIC) study demonstrated that the individual components (both depolarization and repolarization abnormalities) of the Romhilt-Estes LVH score, differ in their prediction of cardiovascular events. Therefore, it is crucial to detect these two phenotypes (voltage or depolarization and repolarization Echo-LVH) [37].

Also, the C5.0 includes the PPK QRS in aVF (to decrease misclassification), and PPK QRS in aVL (to detect Echo-LVH) instead of analyzing the R wave or S wave in an isolated manner as state-of-the-art criteria do. The IHD subgroup could explain this difference. In IHD, the depolarization vector can shift and electrical waves may disappear (i.e., a false negative in Cornell criteria caused by a myocardial infarction manifested as a QS complex in the aVL lead); yet, the PPK QRS persists [24]. Therefore, a PPK QRS should be helpful in patients with known ischemic findings on Echo.

Other ML approaches have been superior in terms of diagnostic performance, than the CHCM. For example, a study that included 4353 LVH patients and 16,933 controls used ensemble, convolutional and deep neural networks to detect LVH, reaching an accuracy, sensitivity, and specificity of 81%-87%, 40%-49%, and 93.4%, respectively. Another study that included 4714 subjects used Bayesian additive regression trees to detect LVH, and reached a sensitivity of 29% and a specificity of 94.6%. However, convolutional neural networks and Bayesian trees are examples of black-box algorithms, and therefore, clinicians would be unaware of which ECG parameter classifies the patient as Echo-LVH positive. This loss of information limits its clinical applicability [21–23]. To overcome this, we decided to model with the C5.0. We previously applied the C5.0 ML algorithm to ECG manual readings and created the *Cardiac hypertrophy model*. This criterion uses five ECG parameters: S wave voltage in lead V4, ST abnormalities, left atrial enlargement, and R wave voltage in aVR. Note that this criterion uses atrial information. This approach led to better diagnostic accuracy (73.3% vs 70.5%), sensitivity (81.6% vs 74.3%), specificity (69.3% vs 68.7%), PPV (56.3% vs 53.8%) and NPV (88.6% vs 84.5%) than the new CHCM [26]. However, the CHCM is half the size of *the Cardiac hypertrophy model*, making it a lot simpler and easier to use [26]. Also, the CHCM is computer-based, so we eliminated the human bias, and inter- and intra-observer variability associated with manual readings, leading to more reproducible results.

## Limitations and future studies

As the study is retrospective, it can harbor selection bias, and although we included a representative sample from our hospital, many patients had a high cardiovascular risk so we cannot generalize the results to the low cardiovascular risk population. Nevertheless, this proof-of-concept study can contribute to finding other ECG parameters capable of detecting Echo-LVH. Also, we used the Mosteller formula to obtain the BSA with the indexed LVM, which is not optimal in the setting of this obese population. The next study will search for the best indexation method to maximize the ECG's diagnostic sensitivity for Eco-LVH.

Of note, the CHCM uses predictors of Echo-LVH but no ECG predictors of mortality; whether the capacity of the predictors included in the CHCM model to predict mortality independently of mortality prediction by the Echo-LVH itself, remains unanswered, and future studies should address this issue. The CHCM was modestly accurate (70.5%) in 439 selected patients, and in a smaller validation cohort, the CHCM´s accuracy was lower (63.5%). The C5.0 algorithm is sensitive to unbalanced training/validation data, explaining the loss of accuracy in the external validation group. The external validation group had a smaller proportion of patients with Echo IHD. However, using this methodology in more extensive and more balanced datasets could yield better results.

## Conclusion

In conclusion, ECG computer-based data (Philips DXL-16 algorithm) and the C5.0 ML algorithm found a new set of ECG parameters that predict Echo-LVH in our population. The *CHCM* is simple and easy to understand and, therefore, we believe it will be applicable in daily clinical practice. The CHCM detects two electrical LVH phenotypes: LVH with repolarization abnormalities or LVH with increased voltage. The CHCM has a similar accuracy, and slightly more sensitivity than the state-of-the-art criteria. A study with a more significant sample could prospectively validate the applicability of the CHCM. Also, other studies should address whether the CHCM can predict cardiovascular morbidity and mortality.

## Supporting information

**S1 Table. ECG parameters analyzed by the Philips DXL-16 algorithm.** This table shows a comprehensive explanation of 458 ECG parameters obtained with the Philips DXL-16 algorithm. This high number of ECG parameters allows an extensive analysis of the electricity of the heart. All these measurements are obtained in every lead.
(DOCX)

## Acknowledgments

We are most grateful to Dr. José Luis Assad Morell, Head of the Cardiology Department at *Hospital Christus Muguerza*, for his support in consulting patient records, and Alejandro De la Garza Salazar and Gregorio Garza for the digital art design.

## Author Contributions

**Conceptualization:** Fernando De la Garza Salazar, Maria Elena Romero Ibarguengoitia, Arnulfo González Cantú.

**Data curation:** Fernando De la Garza Salazar, Arnulfo González Cantú.

**Formal analysis:** Fernando De la Garza Salazar, Arnulfo González Cantú.

**Funding acquisition:** José Ramón Azpiri López.

**Investigation:** Fernando De la Garza Salazar.

**Methodology:** Fernando De la Garza Salazar, Maria Elena Romero Ibarguengoitia, Arnulfo González Cantú.

**Project administration:** Fernando De la Garza Salazar.

**Resources:** José Ramón Azpiri López.

**Software:** Fernando De la Garza Salazar, Arnulfo González Cantú.

**Supervision:** Maria Elena Romero Ibarguengoitia, Arnulfo González Cantú.

**Validation:** Fernando De la Garza Salazar, Maria Elena Romero Ibarguengoitia, José Ramón Azpiri López, Arnulfo González Cantú.

**Visualization:** Fernando De la Garza Salazar, Maria Elena Romero Ibarguengoitia, Arnulfo González Cantú.

**Writing – original draft:** Fernando De la Garza Salazar.

**Writing – review & editing:** Fernando De la Garza Salazar, Maria Elena Romero Ibarguengoitia, Arnulfo González Cantú.

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
