## [Decision Letter · Decision Letter 0]

1 Oct 2021

PONE-D-21-23338Optimizing ECG to detect echocardiographic left ventricular hypertrophy using computer-based ECG data and Machine LearningPLOS ONE

Dear Dr. González Cantú,

Thank you for submitting your manuscript to PLOS ONE. After careful consideration, we feel that it has merit but does not fully meet PLOS ONE’s publication criteria as it currently stands. Therefore, we invite you to submit a revised version of the manuscript that addresses the points raised during the review process.

We look forward to receiving your revised manuscript.

Kind regards,

Kazuaki Negishi

Academic Editor

PLOS ONE

Journal Requirements:

2. In ethics statement in the manuscript and in the online submission form, please provide additional information about the patient records used in your retrospective study. Specifically, please ensure that you have discussed whether all data were fully anonymized before you accessed them and/or whether the IRB or ethics committee waived the requirement for informed consent. If patients provided informed written consent to have data from their medical records used in research, please include this information.

3. We note that you have stated that you will provide repository information for your data at acceptance. Should your manuscript be accepted for publication, we will hold it until you provide the relevant accession numbers or DOIs necessary to access your data. If you wish to make changes to your Data Availability statement, please describe these changes in your cover letter and we will update your Data Availability statement to reflect the information you provide

Reviewers' comments:

Reviewer's Responses to Questions

**Comments to the Author**

1. Is the manuscript technically sound, and do the data support the conclusions?

Reviewer #1: Partly

Reviewer #2: Yes

2. Has the statistical analysis been performed appropriately and rigorously? 

Reviewer #1: Yes

Reviewer #2: Yes

3. Have the authors made all data underlying the findings in their manuscript fully available?

Reviewer #1: Yes

Reviewer #2: Yes

4. Is the manuscript presented in an intelligible fashion and written in standard English?

Reviewer #1: No

Reviewer #2: Yes

5. Review Comments to the Author

Reviewer #1: The authors have conducted a useful case control study to validate the CHCM algorithm that they have developed. While it has limitations in its diagnostic accuracy relative to convolutional neural networks, there are advantages to having a simple system that does not have a black box algorithm. The authors should be commended for their work in optimising the CHCM model. I have a few revision suggestions which related to methodology and results. There are also some grammatical errors/clarifications which I have noted. I have included these as minor revisions.

Major Content Queries

There is now substantial evidence indicating the pitfalls of using BSA alone in indexing LVM. Did you consider using other indexation methods, such as height2.7 (De Simone et. al.) or fat free mass? Were alternative indexation methods considered to account for obesity, particularly given both of your population cohorts (LVH negative and LVH positive) were overweight? BSA has been shown to be a poor indexation method in the setting of obesity.

Did you look into the impact of increased body size on ECG voltage criteria, particularly given your population had a mean BMI of 28.7m2?

Line 131 (methods) -> no formula is listed here for body surface area. Which calculation did you use and what was the rationale for choosing this method? It is important to note that formulae such as Du Bois Du Bois are based on a very small cohort of 9 patients over 80 years ago, and as such have been discouraged in more recent literature for indexing body size variables.

Line 233 What was the variance (e.g. standard deviation) that you used in your sample size calculation? What outcome measure is the delta of o.1 referring to?

Were you able to explore whether certain patient characteristics affected your CHCM model’s accuracy, sensitivity or specificity? For example, was CHCM better/worse based on gender, BMI or the presence of IHD?

Minor grammatical/syntax queries

Line 21 (Abstract) -> Rather than “(ECHO-LVH) produces cardiovascular mortality”, the authors may consider rewording this e.g. “independent predictor of mortality”

Line 85 (background) -> change “produces” to “produce”

Line 88 -> change “are exploring” to “have explored”

Line 204 -> change “de” to “the”

Line 211 -> change “inhomogeneity” to “in homogeneity”

Line 213-> change “cut-values” to “cut-off values”

Lines 216/217 -> this sentence is difficult to understand, suggest breaking into smaller sentences “Also, decision trees can continue to grow indefinitely as more and more predictors are presented to the C5.0 algorithm -this is called overfitting- and created precise predictions in new datasets”

Reviewer #2: This paper presents a method that detects echocardiographic left ventricular hypertrophy (Echo-LVH) using electrocardiogram (ECG). The authors used the Philips DXL-16 algorithm to extract 458 ECG parameters. They then used the C5.0 machine learning (ML) algorithm to automatically find the most relevant parameters associated with Echo-LVH and used them to train a predictive model. The proposed model was evaluated in a retrospective manner on 439 patients and compared with other conventional detection approaches. The results show that the proposed predictive model had similar accuracy to other approaches using a new and fewer sets of parameters.

The proposed cardiac hypertrophy computer-based model (CHCM) is an extension of the authors’ prior work, Cardiac Hypertrophy Model which uses manually selected ECG parameters including ST ECH-LVH, voltage-left atrial enlargement LVH and voltage-duration LVH coupled with C5.50 ML algorithm. To me, it was difficult to clearly understand the contribution of CHCM in comparison to existing models. Apart from being able to leverage the digital data extracted from Phillips DXL-16 algorithm, there seems to be limited new insights and findings introduced from this paper. The in-depth discussions about derived ECG parameters and their corresponding roles in detecting Echo-LVH in comparison to existing algorithms would be important.

Other detailed comments:

1. Page 6, line 101: Inform  Informed

2. Page 6, line 108: STARD  not defined

3. Page 6, line 109: ML  Machine Learning (ML)

4. Page 9, line 172: most of the sample  what is sample in this context?

5. Page 13, line 244-249: The sentences in this paragraph are redundant.

6. Page 15, Table 2: Total sample  Total Samples

7. Page 15, Table 2: why is the number of total samples 321? Why not 439, consistent with Table 1?

8. Page 17, line 309: more accuracy  higher accuracy

9. Page 23, line 398: ‘However, the CHCM is half the size of the Cardiac Hypertrophy model, which gives a more biological coherent model based on the principle of parsimony-the simpler explanation is the most probable’  this seems to be an over justification that explains nothing much. Please be specific and articulate the differences.

6. PLOS authors have the option to publish the peer review history of their article (what does this mean?). If published, this will include your full peer review and any attached files.

Reviewer #1: No

Reviewer #2: No

---

## [Author Response · Author response to Decision Letter 0]

4 Oct 2021

Thank you for dedicating time to this study. We addressed all the comments.

---

## [Decision Letter · Decision Letter 1]

28 Oct 2021

PONE-D-21-23338R1Optimizing ECG to detect echocardiographic left ventricular hypertrophy using computer-based ECG data and Machine LearningPLOS ONE

Dear Dr. González Cantú,

Thank you for submitting your manuscript to PLOS ONE. After careful consideration, we feel that it has merit but does not fully meet PLOS ONE’s publication criteria as it currently stands. Therefore, we invite you to submit a revised version of the manuscript that addresses the points raised during the review process.

We look forward to receiving your revised manuscript.

Kind regards,

Kazuaki Negishi

Academic Editor

PLOS ONE

Journal Requirements:

Additional Editor Comments:

Thank you for amending your manuscript. I believe that the manuscript became much stronger. However, as one of the reviewers pointed out, we would recommend seeking assistance from a native English speaker with scientific writing skills in order to help our readers.

Reviewers' comments:

Reviewer's Responses to Questions

**Comments to the Author**

1. If the authors have adequately addressed your comments raised in a previous round of review and you feel that this manuscript is now acceptable for publication, you may indicate that here to bypass the “Comments to the Author” section, enter your conflict of interest statement in the “Confidential to Editor” section, and submit your "Accept" recommendation.

Reviewer #1: (No Response)

Reviewer #2: All comments have been addressed

2. Is the manuscript technically sound, and do the data support the conclusions?

Reviewer #1: Yes

Reviewer #2: Yes

3. Has the statistical analysis been performed appropriately and rigorously? 

Reviewer #1: Yes

Reviewer #2: Yes

4. Have the authors made all data underlying the findings in their manuscript fully available?

Reviewer #1: Yes

Reviewer #2: Yes

5. Is the manuscript presented in an intelligible fashion and written in standard English?

Reviewer #1: No

Reviewer #2: Yes

6. Review Comments to the Author

Reviewer #1: The authors have made appropriate major changes, however none of the extensive grammatical issues identified have been addressed. As a result, the paper is not written in standard english. I would suggest that the authors seek assistance from an English grammer expert to proof read their manuscript prior to resubmission.

Reviewer #2: The authors have addressed all the comments satisfactorily. The contributions of the paper have been articulated.

7. PLOS authors have the option to publish the peer review history of their article (what does this mean?). If published, this will include your full peer review and any attached files.

Reviewer #1: No

Reviewer #2: No

---

## [Author Response · Author response to Decision Letter 1]

1 Nov 2021

Reviewer #1: The authors have made appropriate major changes, however none of the extensive grammatical issues identified have been addressed. As a result, the paper is not written in standard english. I would suggest that the authors seek assistance from an English grammer expert to proof read their manuscript prior to resubmission.

Response: We sought assistance from an English grammar expert, and now the proof is written in standard English. Thank you.

Reviewer #2: The authors have addressed all the comments satisfactorily. The contributions of the paper have been articulated.

Response: Thanks.

---

## [Editor Report · Decision Letter 2]

15 Nov 2021

Optimizing ECG to detect echocardiographic left ventricular hypertrophy with computer-based ECG data and Machine Learning

PONE-D-21-23338R2

Dear Dr. González Cantú,

We’re pleased to inform you that your manuscript has been judged scientifically suitable for publication and will be formally accepted for publication once it meets all outstanding technical requirements.

Kind regards,

Kazuaki Negishi

Academic Editor

PLOS ONE

Additional Editor Comments (optional):

Thank you for revising your manuscript. The current form reads much better.
---

## [Editor Report · Acceptance letter]

17 Nov 2021

PONE-D-21-23338R2 

Optimizing ECG to detect echocardiographic left ventricular hypertrophy with computer-based ECG data and Machine Learning. 

Dear Dr. González Cantú:

I'm pleased to inform you that your manuscript has been deemed suitable for publication in PLOS ONE. Congratulations! Your manuscript is now with our production department. 

Kind regards, 

on behalf of

Dr. Kazuaki Negishi 

Academic Editor

PLOS ONE